



# Surfaces of Silver Birch (*Betula pendula*) are Sources of Biological Ice Nuclei: *In-vivo* and *In-situ* Investigations

Teresa M. Seifried[1], Paul Bieber[1], Laura Felgitsch[1], Julian Vlasich[1], Florian Reyzek[1], David G. Schmale III[2], Hinrich Grothe[1]

[1]Institute of Materials Chemistry, TU Vienna, Vienna, 1060, Austria
[2]School of Plant and Environmental Sciences, Virginia Tech, 24061-0390 Blacksburg, Virginia, USA

*Correspondence to*: Hinrich Grothe (hinrich.grothe@tuwien.ac.at)

**Abstract.** Silver birch (*Betula pendula*) are known to contain ice-nucleating macromolecules (INMs) to survive in harsh environments. However, little is known about the release and transport of INMs from birch trees into the atmosphere. In this

study, we conducted *in-situ* and *in-vivo* investigations on INMs from nine birches growing in an alpine valley (Ötztal, Austria). A detailed analysis of drill cores showed that INM concentration increases towards outer layers, reaching its maximum near the surface. Aqueous extracts from the surfaces of leaves, bark, primary wood and secondary wood contained INMs (34/36) with concentrations ranging from $9.9 \cdot 10^5$ to $1.8 \cdot 10^9$ INMs cm$^{-2}$. In a field study, we analysed the effect of precipitation on the release of these INMs attached to the surface of the trees. These experiments showed that INMs are splashed and aerosolized

into the environment during rainfall events, at concentrations and freezing temperatures similar to *in-vivo* samples. Our work sheds new light on the release and transport of INMs from birch surfaces into the troposphere. Birches growing in boreal and alpine forests should be considered as an important terrestrial source of INMs.

## 1 Introduction

If temperatures at ambient pressure fall below 0°C, ice is the thermodynamically favourable state of water (Cantrell and

Heymsfield, 2005; Hegg and Baker, 2009; Murray et al., 2010). For the phase transition from liquid to solid state, water molecules need to arrange in an ice like structure. Depending on the temperature the formed ice embryo needs to overcome a critical cluster size to initiate freezing of the bulk, which typically takes place well below 0°C (Cantrell and Heymsfield, 2005; Turnbull and Fisher, 1949). Homogeneous freezing temperatures for small droplets as present in clouds are typically below -35°C (Pruppbacher and Klett, 1997). A broad spectrum of substances has been found to catalyse freezing at higher

temperatures, in a process called heterogeneous ice nucleation (Dorsey, 1948; Murray et al., 2012). Particles that trigger higher freezing temperatures are referred to as ice-nucleating particles (INPs) (Vali et al., 2015). Among these are mineral dusts (Broadley et al., 2012; Zolles et al., 2015), soot (DeMott, 1990; Gorbunov et al., 2001) and biogenic particles (Pöschl et al., 2010), and when airborne, have the potential to affect cloud glaciation and consequently weather and climate (Lohmann, 2002; Mishchenko et al., 1996; Forster et al., 2007; Baker, 1997). The land surface is a major contributor to global aerosols (Fröhlich-

Nowoisky et al., 2016; Pöschl, 2005; Després et al., 2012; Jaenicke, 2005). Atmospheric concentrations of primary biological



aerosol particles (PBAPs) are highly dependent on sampling sites and meteorological factors (Jones and Harrison, 2004). PBAPs include a wide variety of substances with varying sizes from millimetres down to nanometres (e.g. fragments of insects, scurf from plants, pollen, spores, bacteria, viruses, etc.), with cellular material carrying proteins being an important part of it (Jaenicke, 2005). However, our knowledge on the distribution and impact of PBAPs in the atmosphere is still rather limited.

Heterogeneous ice nucleation regarding biological organisms has been reported in plants (Diehl et al., 2001; Pummer et al., 2012), bacteria (Lindow et al., 1982; Wolber et al., 1986), fungi (Pouleur et al., 1992; Fröhlich-Nowoisky et al., 2015; Kunert et al., 2019; Iannone et al., 2011; Haga et al., 2013), moss spores (Weber, 2016), and lichen (Kieft, 1988). Plants growing in temperate environments use mechanisms such as extracellular freezing (Burke et al., 1976; Storey and Storey, 2005; Pearce, 2001) or extra-organ freezing (Quamme, 1978; Ishikawa and Sakai, 1981) to survive sub-zero temperatures.

Consequently, ice nucleation is an important means of regulating freezing in plants. Extracellular freezing in frost-tolerant woody plants occurs mostly in bark tissues (Burke et al., 1976; Ashworth et al., 1988; Ashworth, 1996). INPs trigger ice crystal formation in extracellular spaces that allow the withdrawal of water from the cell. This leads to an increase in the supercooling capacity of the cell, preventing it from damage (Ishikawa and Sakai, 1981). Extra-organ freezing is common in leaf buds (Ishikawa et al., 2015), where bud scales serve as ice sinks, moving water from the flower primordia. The resulting

supercooling of flower primordia prevents the tissues from freezing (Ishikawa et al., 2015).

      Many frost-hardy plants contain INPs. Leaves of winter rye (*Secale cereal*) embody INPs active between -7 and -8°C (Brush et al., 1994). The stems of high-bush blueberries (*Vaccinium corymbosum*) (Kishimoto et al., 2014b), wood of *Prunus* (Gross et al., 1988), and a variety of berries from perennial plants (Felgitsch et al., 2019) are also known to contain INPs, whereby the former show highest INP concentrations in autumn, just before frost events occur (Kishimoto et al., 2014a).

Additionally, decayed leaf litter was found to be a potent source for INPs (Conen et al., 2016; Conen et al., 2017; Schnell and Vali, 1973), some of which may be stable for almost 50 years (Vasebi et al., 2019). Conen et al. (2017) found that the emission of these INPs (active at temperatures above –8°C) increases in autumn and it is hypothesized that the vegetation could contribute to local climate. Moreover, INPs are found to be present in various pollen from different tree species (e.g. birch, pine, juniper, etc.) (Pummer et al., 2012; Diehl et al., 2001). Past studies showed that INPs from biological systems are rather

in a macromolecular size range (ice-nucleating macromolecules, INM) (Pummer et al., 2012; Kunert et al., 2019). If those INMs can be easily washed off a plant's surface, they would have the potential to become aerosolized and serve as an important source for INPs in the atmosphere.

      Heavy rain events and thunderstorms have been associated with high INP concentrations (Bigg and Miles, 1964; Isono and Tanaka, 1966). More recently, studies highlight that rainfall triggers the release of PBAPs into the atmosphere

(Huffman et al., 2013; Rathnayake et al., 2017; Prenni et al., 2013; Hara et al., 2016; Tobo et al., 2013). Huffman and colleagues showed that the burst in PBAP correlates with fluorescence activity and high INP concentrations (Huffman et al., 2013). During rainfall, splash-induced emissions of fungal spores can occur (Hirst, 1953; Allitt, 2000; Kim et al., 2019). Rain droplets impacting on leaf surfaces hosting ice-nucleation active microbes (Morris et al., 2004) or soluble INMs (Pummer et al., 2015) may produce aerosols containing INPs. High relative humidity may also result in an increase in INPs (Wright et al., 2014),



priming fungi to eject spores (Jones and Harrison, 2004) and encouraging pollen grains to germinate or rupture due to enhanced osmotic pressure (Taylor et al., 2004). Both processes release particles with aerodynamic diameters smaller than 2 μm (Grote et al., 2003), with the ability to be transported into the atmosphere by wind (Pöschl, 2005). Pollen grains of birch are known to trigger ice nucleation at around -18°C (Pummer et al., 2012; Diehl et al., 2001). Pummer et al. (2012) were able to show that these ice nuclei are in the macromolecular size range since INMs could pass through a 200 nm cut-off filter whereas the

pollen grains are retained (Pummer et al., 2012; Pummer et al., 2015). The INMs are easily detached from the pollen and are released in highest concentrations to the aqueous phase. INMs are not only present in the pollen grains of birches, but are spread throughout different parts of the tree, including leaves and branch wood (Felgitsch et al., 2018). However, only extracts of pulverized plant material were analysed so far (Felgitsch et al., 2018) and the availability of INMs from birches to the surrounding environment remains unclear. In this paper, we extend the work of Felgitsch et al. (2018) by analysing the

distribution of INMs throughout different birch trees, with a focus on intact surfaces of silver birch trees as sources of biological ice nuclei. We hypothesized that INMs are located on the surface of birches and are transported into the troposphere. The specific objectives of this work were to study the distribution and concentrations of INMs in birch tissues and investigate the transport of INMs from birch trees into the environment. Drill cores from three selected trees were characterized for their ice nucleation activity. INMs were extracted from leaves, branch wood (primary and secondary wood) and bark. A field study was

conducted using four birch trees at different locations to analyse the effect of precipitation on the release of INMs from the surface of the trees. Collectively, our work sheds new light on the release and transport of INMs from the terrestrial biosphere into the atmosphere.

## 2 Methodology

### 2.1 Sample collection

Birch trees were selected in an alpine valley (Ötztal) in the western part of Austria (GPS: 48.756098, 15.891850) (Felgitsch et al., 2018). The valley climbs from 799 m above sea level, where the environment is dominated by fields and forests to 3400 m where the valley ends in a glacier area. This area is remote and sparsely populated; anthropogenic aerosol emissions (e.g. traffic and industrial emission, biomass burning, etc.) are limited. We collected samples from nine different mature birches growing throughout the valley along an altitudinal gradient from 799 to 1925 m (timberline). The higher the altitude the more

the environment is comparable to northern latitudes (boreal forests), where trees need to cope with cold winters. We numbered the sampled trees alphabetically (Tree A to I). Tree A and B are growing at the foot of the valley in a forest. Trees C, D, E, F, H and I are located next to a mountain river (riparian forest). Tree G is not surrounded by other trees and located at the highest point growing directly at the timberline (1925 m). GPS way points, altitudes and further information on the sampled trees are given in Table 1. Figure 1 shows the valley and location of the sampled trees (map adapted from Google Earth©).

We took bark, branches (primary and secondary wood), leaves as well as drill core samples in summer 2016 (see Figure 2). Ice nucleation activity of powder extracts from leaf and branch wood samples has been reported previously



(Felgitsch et al., 2018). However, in this study we focused on the investigation of intact tissues. Primary wood is newly grown and still photosynthetically active. In contrast, secondary wood is older, appears brown and is not photosynthetic active anymore. We took approx. 50 mm sections of these branch wood samples, originating from the same branch per tree. From

the trunk, we collected bark samples using tweezers and drill cores at 1 m height, which were obtained with a pole testing drill (5 mm diameter, Haglöf, Sweden). The drill cores consist of the bark (outermost layer), the bast and the inner wood (see Figure 2, (e)). A microscopic picture of each segment is given in Figure 2 on the bottom ((f), (g), (h)). All tools used during sampling were disinfected with ethanol (approx. 90 vol%) after each usage. Samples were frozen within a few hours of sampling at -20°C.

105        In October, 2019, a series of precipitation collectors were placed underneath the birches Tree E, F, G and H. These rain collectors consisted of sterile centrifuge tubes (polypropylene, 50 mL, Brand®, Germany) mounted on wooden sticks anchored to the ground with three guy-lines (Figure 3). Three to four collectors were placed under each tree and cardinal positions (north (N), east (E), south (S), west (W)) were noted. All collectors were placed directly under the crown of the tree. One collector was set up next to each tree in an open terrain (> 5 m away from the tree) collecting pure rainwater to serve as a

control (blank sample).

### 2.2 Sample preparation

The drill cores of Tree A, C and I were separated in bark, bast, and approx. three 5 mm large sections of inner wood (see Figure 2). To determine the surface availability of each part of the drill core, we approximated the segments as cylinders. Each segment was extracted with ultra-pure water for six hours. Since we were interested in substances in the macromolecular size range,

the coarse fraction of the extracts was removed and the remaining aqueous phase was then centrifuged at 3500 rpm (1123 $xg$) for 10 min and filtered with a 0.2 µm syringe filter (cellulose acetate membrane, sterile, VWR, USA). The surface area of each drill core segment per volume of water used for the extraction is summarized in Table 2.

        The branch wood samples were cut in approx. 10 mm sections, sealing the edges with paraffin wax (Sigma Aldrich, USA) to avoid leakage of components through the cut surface, and extracting the samples with ultra-pure water (between 0.6 and

3 mL) for six hours. The wax was tested for ice nucleation activity prior sample treatment and was found to be inactive. In case of the leaves we used whole leaves, again covering the petiole with paraffin wax and extracting the samples in 5-10 mL water per sample for six hours. The extraction volume varies due to the varying sizes of the samples. The bark was analysed using a 5 mm punch and extracting three punched pieces of bark per sample in 1 mL ultrapure water for six hours. In line with drill core samples, the particular solid wood or leaf sample was removed and the remaining sample solution was centrifuged

(1123 $xg$) and filtered (0.2 µm syringe filter). In order to calculate the extracted samples' surfaces, samples were roughly estimated as geometric figures: branch wood as cylinders, leaves as convex deltoids and bark punches as circles. The detailed calculations are described in chapter 2.5. The surfaces per extracted millilitre are given in Table 3.



Rain samples were stored in a -20°C freezer. The volume of each rain sample was determined after the sample was defrosted, immediately prior to analysis. One millilitre aliquots were taken from these thawed rain samples, filtrated with a 0.2

µm syringe filter, as described for the wood and leaf samples, and measured for ice nucleation activity.

### 2.4. Cryo-microscopy

All freezing experiments (immersion freezing mode) were performed using the cryo-microscopy setup VODCA (Vienna Optical Droplet Crystallization Analyser). A detailed description of the setup is given in Felgitsch et al. (2018). In short, the setup consists of two main components: an incident light microscope (BX51M, Olympus, Japan) with an attached camera

(MDC320, Hengtech, Germany) linked to a computer and the cryo-cell. The cryo-cell is a polymer-based compartment that can be closed airtight. It contains a cooling unit consisting of a Peltier-element (Quick-cool QC-31-1.4-3.7M) with a thermocouple fixed on top and a heat exchanger, cooling the warm side of the Peltier-element during freezing experiments. The samples are all measured as aqueous components of an emulsion created on a clean glass slide which is placed on top of the Peltier-element during the cooling process. A LabView based software enables us to record videos during the freezing

process. All freezing experiments were performed with a cooling rate of 10°C min$^{-1}$. Only droplets in the size range between 15 and 40 µm (droplet volume: 1.8 – 34 pL) were included in our evaluation of the freezing experiments.

### 2.5 Data Analysis

Considering ice nucleation to be a time-independent process, the number of INMs or INPs active above a certain temperature can be expressed by the cumulative nucleus spectrum ($K(T)$) (Murray et al., 2012; Vali, 1971) as stated in Equation (1).


$$K(T) = \frac{D \cdot \ln(1 - f_{ice}(T))}{V_{droplet}} \tag{1}$$

$$\text{with } f_{ice} = \frac{n_{frozen}}{n_{total}}$$

$D$ is the dilution factor, $V_{droplet}$ is the analysed droplet volume (8.2 pL using VODCA) and $f_{ice}(T)$ is the fraction of frozen droplets at a given temperature, whereas $n_{frozen}$ is the number droplets which froze at a regarded temperature and $n_{total}$ is the total number of droplets frozen during an experiment.

To refer the number of INMs extractable from the surface of the respective sample, we modified $K(T)$ by multiplying with the extraction volume, $V_{extraction}$ divided by the surface of the sample, $\sigma_{sample}$ (Equation (2)).

$$K(T) = \frac{D \cdot \ln(1 - f_{ice}(T))}{V_{droplet}} \cdot \frac{V_{extraction}}{\sigma_{sample}} \tag{2}$$





To estimate the area of the sample surfaces, we used approximations as indicated in section 2.2. Drill core segments as well as
primary and secondary wood samples were estimated as cylinders. We used the whole surface area for the core samples and
the surface area without the base for the secondary and primary wood samples (since the top and bottom base were covered
with paraffin wax). Leaves were approximated with the area of two triangles to best capture the kite-like shape (convex
deltoid). In our surface estimations, we accounted the dorsal and ventral side for the area of the leaf. For the bark samples, we
used two punches with a diameter of 5 mm, therefore the surface was approximated as the circular area multiplied by two to
capture the top and bottom of the punch. We did not account for surface roughness. Surface sites are given in Table 2 and
Table 3.

To calculate the INM concentrations extracted from the birches during rainfall events, $K(T)$ from Equation (1) was
modified by multiplying with the rain volume, $V_{rain}$ divided by the area of the precipitation collectors' inlet, $\sigma_{inlet}$ (circular)
(Equation (3)).


$$K(T) = \frac{D \cdot \ln\left(1 - f_{ice}(T)\right)}{V_{droplets}} \cdot \frac{V_{rain}}{\sigma_{inlet}} \tag{3}$$

Calculating $K(-34°C)$ in dependency of the sampled rain volume per surface (cross-section area of sample collector) made it
possible to compare the data of rain samples with samples extracted in the laboratory. In immersion freezing experiments, only
INMs with the highest activity present in a droplet can be observed and INMs in the same droplet with lower activity are not
captured. Thus, INMs active at lower temperatures are easily underrepresented in samples with high concentrations of INMs.
To avoid underestimation, a sample was diluted and re-measured when it froze purely heterogeneously. Thus, the full range
of present INMs is captured within the experiments.

**3 Results**

**3.1 Drill Cores (*in-vivo*)**

The results of drill cores show that $K(-34 °C)$ values are highest at the outermost parts of the tree, meaning the bark and the
bast (Figure 4). $K(-34°C)$ gives the number of INMs active at temperatures higher than -34°C, attributing purely to
heterogeneous freezing events (calculated by Equation (2)). The bark values varied between $1.8 \cdot 10^7$ cm$^{-2}$ (Tree A) and $9.8 \cdot 10^7$
cm$^{-2}$ (Tree C). Yet, the bast sample from Tree A did not freeze heterogeneously at all, likely due to a concentration of INMs
under the limit of detection for the freezing assay. However, the Tree C bast sample showed the highest $K(-34°C)$ value ($4.2 \cdot 10^8$
cm$^{-2}$) of all analysed core samples. The inner wood segments of all three drill cores exhibited significantly lower INM
concentrations compared to the rest, ranging between $1.7 \cdot 10^6$ cm$^{-2}$ (Tree A, inner wood segment 2) and $6.7 \cdot 10^7$ cm$^{-2}$ (Tree A,
inner wood segment 3). To model the concentration gradient of INMs throughout the drill cores, the data was fitted with an



exponential function (y=4.09·10$^7$·e$^{-1.20x}$). We see that the INM concentration gradient within the trunk of the tree increases towards outer layers of the trunk, reaching the maximum near the surface.

### 3.2. Barks, leaves, branch woods (*in-vivo*)

Nearly all analysed surface extracts of bark, leave and branch wood samples released INMs with *K(-34°C)* values in the order of magnitude between 10$^5$ and 10$^9$ cm$^{-2}$. Figure 5 provides *K(-34°C)* values as well as the cumulative nucleus spectrum as a function of temperature (*K(T)*) providing information on the course of heterogeneous freezing including the onset freezing temperatures ($T_{on}$). In general, we observed rather high concentrations for bark and branch wood samples compared to leaves. Two of all analysed samples did not show any ice nucleation activity, both of which were primary wood, one from Tree F and another one from Tree H. Further, $T_{on}$ values across all heterogeneously nucleated samples ranged from -14.0°C (Tree B, secondary wood) to -28.6°C (Tree H, bark). Bark, leaves, primary and secondary wood samples from Tree A, C, D and G showed similar $T_{on}$ values. However, $T_{on}$ varies strongly across samples from Tree B, E, F, H and I. Considering the whole data set, the type of sample that started to freeze at the highest temperature varied across the trees. Focusing on all leaf samples, a strong variation of $T_{on}$ is observed ranging from -15.7 (Tree C) to -27.5°C (Tree E). In contrast, secondary woods showed low variations not only for $T_{on}$ but also for *K(-34°C)* values. In addition, secondary wood tended to exhibit highest $T_{on}$ values.

### 3.3. Rainfall event (in-situ)

Ice nucleation data of rain collected underneath the birches reveal heterogeneous freezing in all samples (Figure 6), whereat blank samples did not show heterogeneous freezing at all. *K(-34°C)* values, calculated using Equation (3), varied in the order of magnitude between 10$^6$ and 10$^9$ per cm$^2$ (i.e. 10$^6$ to 10$^9$ INMs were extracted per cm$^2$ area of rain). Within the samples from the same tree *K(-34°C)* values are considerably similar. However, the concentration of sample W (west) from Tree G is approximately two orders of magnitude below the other two samples (S (south), NE (north-east)). Furthermore, $T_{on}$ within the samples from each tree were comparably similar. Nonetheless, $T_{on}$ of sample W from Tree G is an outliner. Moreover, two trees, namely Tree E and F were covered with leaves (most of which were yellow) whereas Tree G and H were leafless. On the one hand, all samples from leaf-covered trees exhibit similar *K(-34°C)* values (higher than 10$^7$ INM cm$^{-2}$). On the other hand, samples from trees without leaves show a high variation in *K(-34°C)* values. Furthermore, the cumulative nucleus spectra indicate steep increases at the beginning of the curves at temperatures between -16.0°C and -23.0°C. $T_{on}$ of rain samples varied between -15.6 and -23.8°C, rather similar to *in-vivo* samples.

### 4 Discussion

The influence of PBAPs acting as INPs in atmospheric processes as well as the transport of these particles from the terrestrial surface to the atmosphere remains elusive. It is well known that many plants growing in boreal and alpine forests contain INPs or INMs to survive in extreme conditions (Sakai and Larcher, 1987). However, less is known about the amount of INMs on



the land surface and actual emission rates. This study shows that INMs from birches are concentrated near the surface of the tree, especially around the trunk. Within the outer layer of the trunk, the water transport and thus the provision of nutrients is

taking place, whereas the inner wood is rather important for the static support. In the bark, extra-cellular freezing mechanisms, where INPs are important, are taking place (Ashworth, 1996) and thus, explain the observation of INMs to be present more intensively on the outer layer of the trunk.

INMs were extracted from all surfaces of leaf and branch wood samples, except for two, both of which were primary wood (originated from Tree H and F). The non-active samples indicate an absence of INMs in the observed droplets of freezing

experiments. This could be caused by the hydrophobic surfaces of primary wood, leading to an INM concentration too low to be captured with our freezing assay. Felgitsch et al. (2018) analysed the INM concentration of powder extracts from birch leaves and branch woods from the same trees analysed within this work. To compare the data from this study to the previous one, we converted the cumulative nucleus concentration from cm$^{-2}$ (which relates to the surface of the sample) to mg$^{-1}$ (see Supplement, Figure 7). The comparison of the $K(-34°C)$ values show that the INM concentration of secondary wood is about

2-3 orders of magnitudes lower for surface extracts than for powder extracts. The same trend can be observed for primary wood, however less pronounced (decrease between 1-2 orders of magnitude). In contrast to the branch wood samples, the INM concentrations of leaf powder and surface extracts were greatly similar, as the surface to mass ratio for powder leaf samples is quite similar to intact leaf samples. Concerning $T_{on}$ of analysed samples, there is a strong agreement with results from birch pollen (Pummer et al., 2012) and pulverized extracts of birch tree tissues (Felgitsch et al., 2018). These results suggest that

birch tissues are a source of a large proportion of the INMs in environment, and the phytobiome on the surfaces of birch trees (including ice-nucleating bacteria) is a minor contributor.

A further finding is the ability of INMs to be extracted by rain from the surface of birches. In all precipitation collectors underneath the studied trees (which were Tree E, F, G, H for the rain samples) INMs were found to be present. The $K(-34°C)$ values of sample W from Tree G is approximately two orders of magnitude below the other two samples from this

tree (S, NE). This could be either due to the sample collector been situated too far from the tree, due to the interaction between rain and the tree's surface being insufficient, or that the western part of the tree exhibits fewer INMs. Two of the trees were covered with leaves (Tree E and F), whereas the other ones were leafless (Tree G and H). Rainwater samples from trees with leaves show low variability, whereat the concentration of samples from leafless trees varies between $1.3 \cdot 10^7$ to $2.6 \cdot 10^9$ INM cm$^{-2}$. Thus, we assume the interaction of rain droplets with the birches surface to be distributed more randomly, when no leaves

are present. However, we see for two out of six samples rather high INM concentrations and therefore leaves are not indispensably for rainfall events to extract ice nucleation material from silver birches.

In our data, we found the highest freezing onset temperature on the surface of intact leaves to be -15.7°C which rather excludes the presence of highly active ice-nucleating bacteria. Furthermore, we observed that INMs can be released into the environment. In 1972, Schnell and Vali discovered leaf litter to contain INPs (Schnell and Vali, 1973) which was later

associated with the presence of the bacteria *Pseudomonas syringae* (typically active above -10°C). Furthermore, Conen et al. (2016) analysed leaf litter on the ground and found concentration to be $2 \cdot 10^2$ INPs µg$^{-1}$ (active above -15°C). In addition, they





studied the transport of these INPs and claimed that the vegetation change in autumn due to decaying leaf litter leads to an increase in atmospheric INP concentrations (Conen et al., 2016). One theory on the transport mechanism could be that INMs situated e.g. on the wet surface of leaves are ejected mechanically by leaf movement caused by the rain, i.e. a mechanism

similar to the bioaerosol generation described by Joung et al. (2017). A rain droplet hits the INM containing surface, small bubbles possibly containing INMs are ejected, which can be transported further (Joung et al., 2017). It is also possible that INMs are released from birch surfaces by microscopic tornadoes generated by raindrop impacts, similar to what has been observed for spores of a fungal plant pathogen (Kim et al., 2019).

If we assume that *Betula pendula* and *Betula pubescens*, which are the main birch species found in Northern Europe

(Beck et al., 2016), behave similarly concerning INA and that the INA throughout the crown of the tree stays comparable, we can estimate the potential role of birches in environmental INM concentrations in Europe. The leaf area index LAI describes the leaf area to ground area ratio in a forest and varies depending on the analysed forest and the method of determination. The LAI of selected studies for northern birch forests and birch tree stands varies between 0.66 and 4.09 $m^2$ $m^{-2}$ (Karlsson et al., 2005; Dahlberg et al., 2004; Heiskanen, 2006; Johansson, 2000; Uri et al., 2007). For our calculations we assumed the LAI to

be 2 $m^2$ $m^{-2}$. Assuming that the birches in northern birch forests behave similarly to our measured trees and that leaves in the upper canopy are comparable to those in the lower canopy, the aqueously extractable INM fraction is $1.99 \cdot 10^{10}$ to $2.68 \cdot 10^{12}$ INM per $m^2$ forest for the leaves alone. Estimating the general leaf area of a tree however is extremely inaccurate since density of a crown is highly dependent on the stand density (Hynynen et al., 2009). To estimate the surface of the trunk, we took the data from investigations of *Betula pendula* stands containing trees between 7 and 60 years of age and field data from a boreal

forest. The data showed a height between 3.2 and 28 m, and diameters between 32 and 640 mm (Hynynen et al., 2009; Johansson, 2000; Ene et al., 2012). Taking these values and approximating the trunk of the tree as a cone, we can estimate the surface area of the stem leading to a surface area between 0.162 and 28.5 m². Estimating the surface of the trunk to be in average 14.3 $m^2$, this adds another $5.7 \cdot 10^{12}$ to $2.6 \ 10^{14}$ INM for the bark on the stem per birch tree. Taking the data from two multiple birch stand investigations the density varied between 0.04 and 10 trees per $m^2$ (Johansson, 2000; Hynynen et al.,

2009; Uri et al., 2012). Leading to $2.8 \cdot 10^{13}$ to $1.3 \cdot 10^{15}$ INM per $m^2$ of the birch stands. This calculation contains an overestimation of the maximum tree surface per $m^2$ since the densest stands are typically young stands with rather thin stems. However, comparing these calculations with the INMs concentration found to be released for one tree after a rain shower (Figure 6), we find the concentrations to be comparable in the order of magnitude. Further, considering a 10 ha birch forest in Northern Europe $2.8 \cdot 10^{18}$ to $1.3 \cdot 10^{20}$ INMs could be released after a six hour rain fall.

Indeed, the findings of our study show that INMs are released during rain events. However, possible transport mechanisms of INMs between the land surface and the atmosphere remains unclear and needs further investigations. In the future, we plan to analyse the release and transport of INMs from birches into the environment using remote controlled vehicles. Very recently we reported first results of a drone-based sampling platform that will be used to analyse airborne INMs above emission sources (e.g. birch forest) (Bieber et al., 2020). INMs are in submicron size range (<200 nm). Aerosolized,



they could be transported over great distances and hence affect atmospheric processes, e.g. cloud glaciation, which further affects precipitation formation and thus influences the hydrological cycle.

**5 Conclusion**

Results from this study shed new light on possible pathways of INMs between the terrestrial biosphere and the atmosphere. We found high concentrations of INMs on the surface of birches. Nearly all biological surfaces (34 out of 36) contained

extractable INMs smaller than 0.2 µm, active at about -20°C. Drill cores of the trunk point out the INMs to be enriched on the surface. We calculated and measured INMs to be present in orders of magnitude of about $10^{13}$ to $10^{15}$ INM per $m^2$ birch forest. Indeed, we suggest that rain induced aerosolization of INMs (i.e. splashing of rain droplets on the tree's surface; soil particles serving as atmospheric shuttles of impacted INMs) contributes as a large source of biogenic INPs beside pollination season and fungal spores. Highlighting the possible pathways of INMs to be transported into the atmosphere during rainfall.

Quantitative determination of these emission fluxes could improve scientific knowledge on aerosol-cloud interactions. We found high similarities between *in-vivo* prepared extracts and *in-situ* sampled rainwater, confirming the hypothesis that rainfall washes of INMs from birch surfaces. However, chemical identification of INMs and airborne measurements are from high importance for further studies. Both of which lead to vertical profiles of INMs above emission sources. Thus, weather and climate models can be adapted to the influence of bioaerosols from alpine or boreal forests.


**Data availability**. All data are available from the corresponding author upon request.

**Author contributions**. TMS, PB, LF, DGS III and HG designed the experiments. JV, FR, TMS and PB performed the experiments. TMS and PB conducted the field measurement. TMS, PB and HG discussed the results. TMS, LF and PB wrote

the manuscript with contributions of all co-authors.

**Competing interests**. The authors declare that they have no conflict of interest

**Acknowledgements.** The authors would like to thank the FWF (Austrian Science Fund, project no. P 26040) for funding and

Romana Bieber for her support during the planning phase of the field campaign.



**Table 1:** Information of the nine sampled birches in Tyrol, Austria. Cores and samples for surface extraction experiments were taken in summer 2016. Rain samples were collected in October 2019 underneath Tree E, F, G and H.

| Sample ID | Sampling Date Surface Extracts | Sampling Date Rain | GPS Waypoints | Altitude [m] | Circumference of trunk at 1 m height[cm] |
|---|---|---|---|---|---|
| Tree A | 07/05/2016 | - | 47.214241, 10.798765 | 799 | 113 |
| Tree B | 07/05/2016 | - | 47.221615, 10.829835 | 799 | 54 |
| Tree C | 07/05/2016 | - | 47.186231, 10.908341 | 851 | 75 |
| Tree D | 07/05/2016 | - | 47.185387, 10.909587 | 851 | 35 |
| Tree E | 07/05/2016 | 29/10/2019 | 46.973163, 11.010921 | 1343 | 96 |
| Tree F | 07/05/2016 | 29/10/2019 | 46.974588, 11.011463 | 1343 | 61 |
| Tree G | 07/07/2016 | 29/10/2019 | 46.878959, 11.024441 | 1925 | 67 |
| Tree H | 07/08/2016 | 29/10/2019 | 46.873275, 11.026616 | 1883 | 36 |
| Tree I | 07/08/2016 | - | 46.873279, 11.026736 | 1883 | 59 |


**Table 2:** Surface area per millilitre of extracted water (ultra-pure) for drill core samples from Tree A, C and I. Bark and bast segments are the outermost layers, followed by three layers of the inner wood (1-3).

| Tree | Bark [cm² mL⁻¹] | Bast [cm² mL⁻¹] | Inner wood segment 1 [cm² mL⁻¹] | Inner wood segment 2 [cm² mL⁻¹] | Inner wood segment 2 [cm² mL⁻¹] |
|---|---|---|---|---|---|
| A | 1,7 | 0,8 | 1,1 | 1,0 | 1,1 |
| C | 0,8 | 0,9 | 0,9 | 0,9 | 1,1 |
| I | 0,4 | 0,8 | 1,2 | 1,2 | 1,0 |


**Table 3:** Surface area per millilitre of extracted water (ultra-pure) for bark, primary and secondary wood, as well as leaf samples.

| Tree | Bark [cm² mL⁻¹] | Prim. wood [cm² mL⁻¹] | Sec. wood [cm² mL⁻¹] | Leaves [cm² mL⁻¹] |
|---|---|---|---|---|
| A | 1.2 | 0,5 | 2,0 | 1,6 |
| B | 1.2 | 0,4 | 1,6 | 2,0 |
| C | 1.2 | 0,6 | 0,8 | 4,1 |
| D | 1.2 | 0,3 | 0,5 | 1,2 |
| E | 1.2 | 0,5 | 0,4 | 1,2 |
| F | 1.2 | 0,4 | 0,9 | 0,9 |
| G | 1.2 | 0,9 | 1,8 | 2,8 |
| H | 1.2 | 0,3 | 1,2 | 2,2 |
| I | 1.2 | 0,5 | 1,6 | 2,2 |



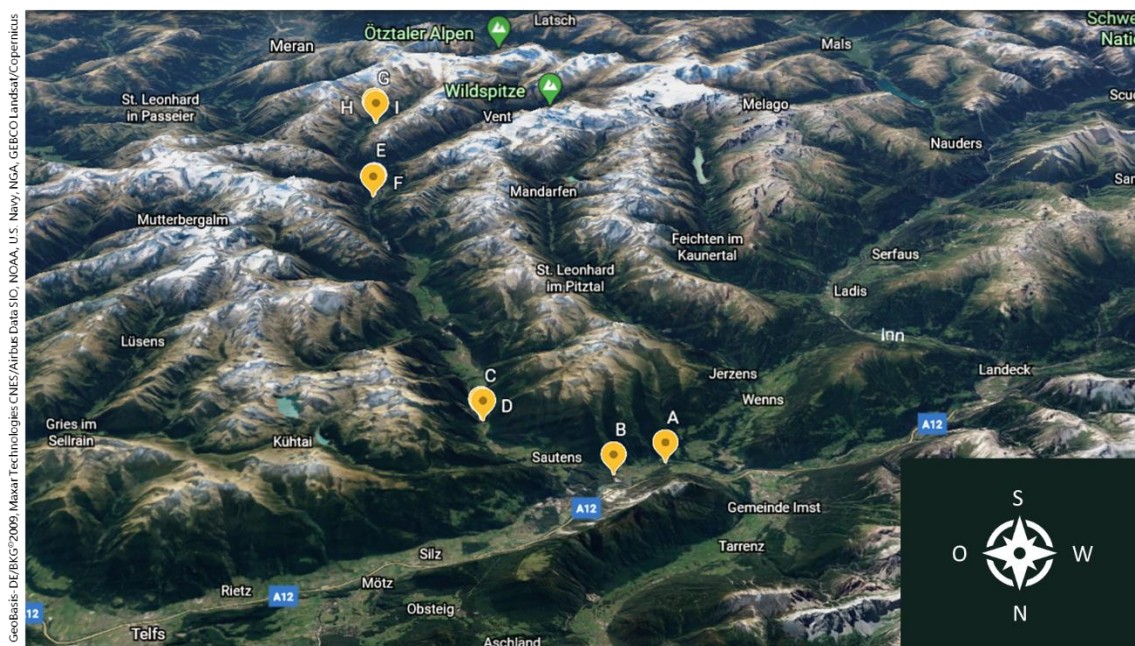

**Figure 1:** Locations of the sampled birches (Tree A to I) along the Ötztal valley in Tyrol, Austria. Picture adapted from Google Earth©, 320 https://earth.google.com/ access: 10/02/2020.

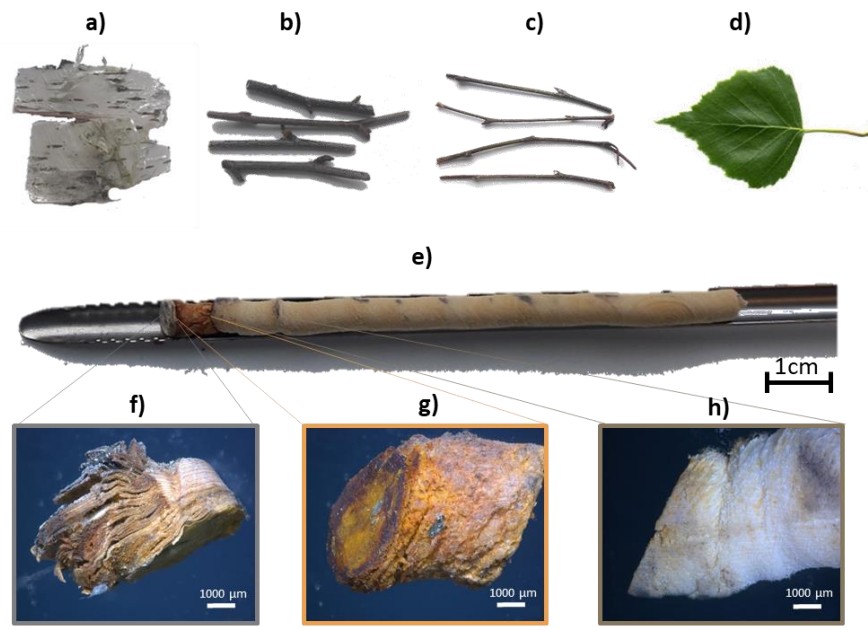

**Figure 2:** Type of samples collected. Top: a) bark, b) secondary wood, c) primary wood, d) leaves. In the middle: e) drill core. Bottom: microscopic picture of f) bark, g) bast and i) inner wood from the drill core (Tree C).






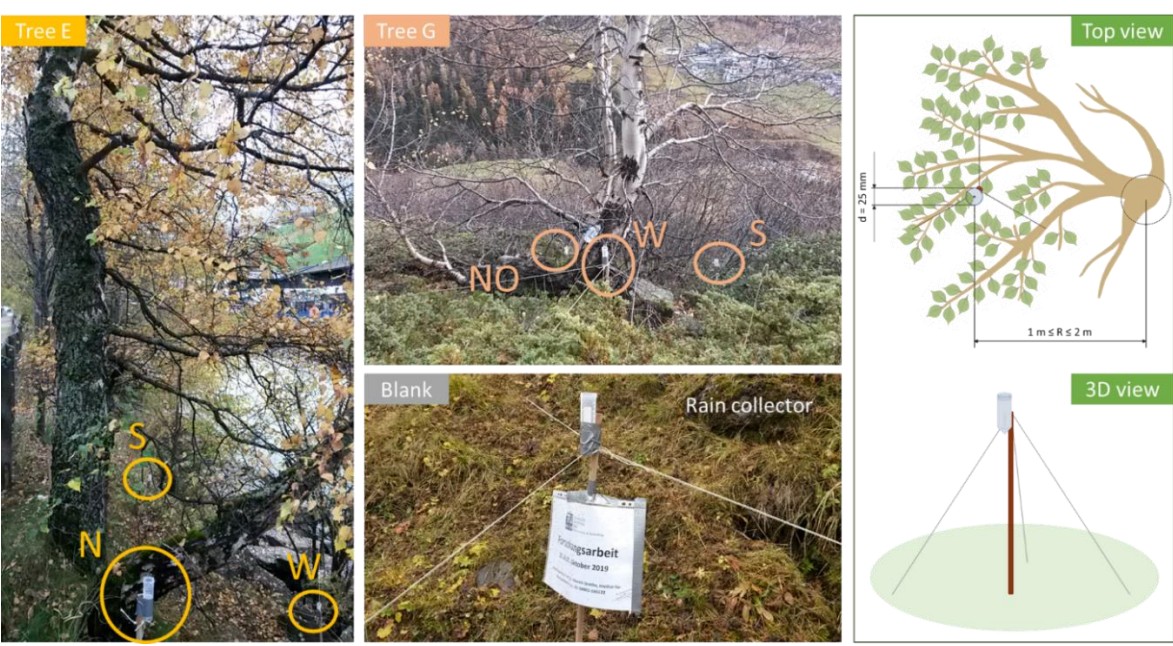

**Figure 3:** Rain sampling set-up underneath Tree E (left, yellow) and Tree G (top middle, light orange). Underneath every sampled tree (Tree E, F, G and H) were three to four rainwater collectors positioned. Next to the trees of interest, we mounted a blank collector, which was placed in an open terrain (bottom middle, grey). Top view as well as a schematic 3D picture of the rain collector (right, green).

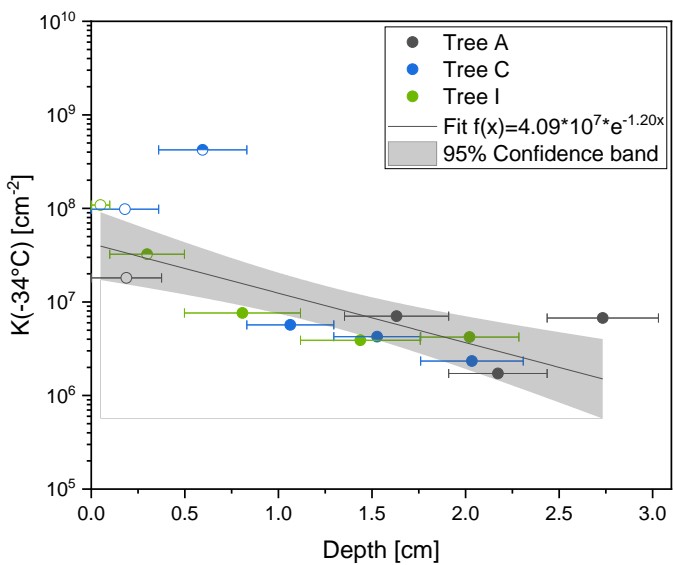


**Figure 4:** Cumulative nucleus concentration at -34 °C ($K(-34°C)$) of the analysed core samples of Tree A (grey), Tree C (blue) and Tree I (green). Bark samples are marked with a hollow symbol, bast samples are marked with a half-filled symbol. Filled symbols correspond to inner wood samples. To assign the depth of the different core segments, the centre of the segments was used. The sizes of the segments are visualized with corresponding horizontal lines. INM concentration increases towards outer layers of the trunk. The bast sample from Tree A does not freeze heterogeneously.




**Figure 5:** Cumulative nucleus spectra *K(T)* of aqueous extracted bark, branch wood and leaf surfaces. Samples originate from nine different birches (named Tree A to I) in Tyrol, Austria. Rectangles show bark samples, dots represent leaves, triangles with the cone end up illustrate primary wood and triangles with the cone end down secondary wood. Nearly all sample freeze heterogeneously. Two samples are not active, both of which are primary wood (Tree F and H).



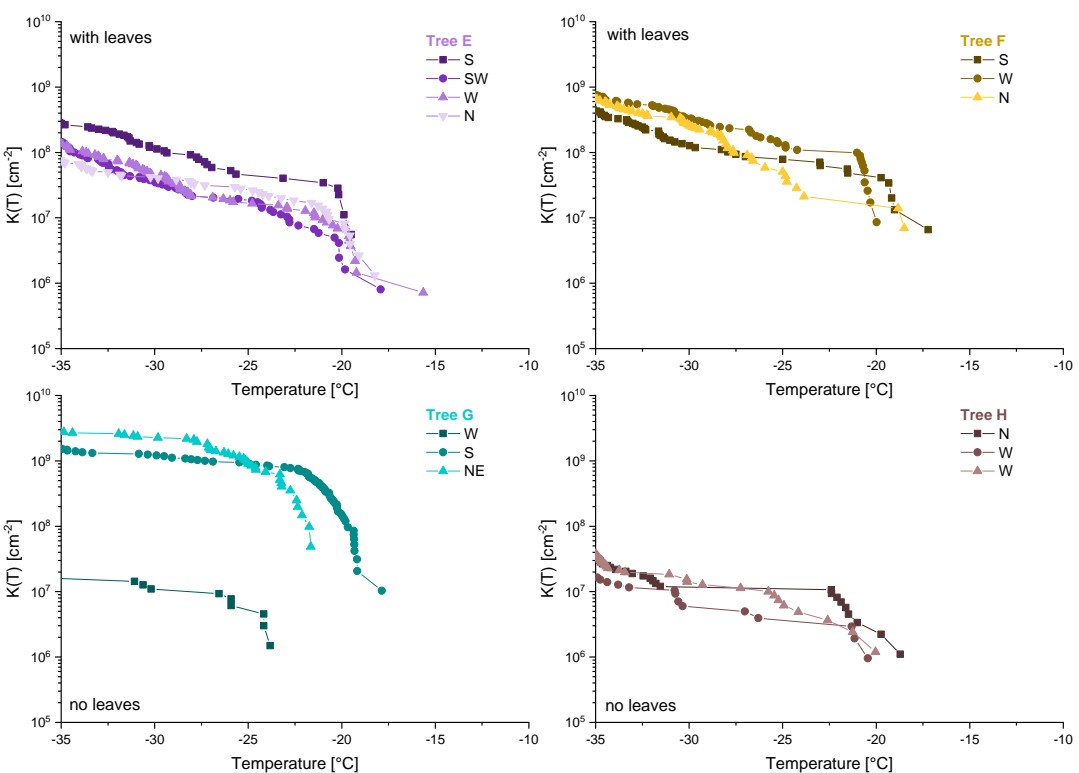

**Figure 6:** Cumulative nucleus spectra *K(T)* of collected rain samples underneath Tree E, F, G and H. Tree E and F were covered with leaves. Tree G and H were leafless. All rain samples collected are ice nucleation active.

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
