# Peer review of "Surfaces of Silver Birch (*Betula pendula*) are Sources of Biological Ice Nuclei: *In-vivo* and *In-situ* Investigations"

_Biogeosciences, 2020_

## Referee Comment (RC1) · Cindy Morris (Referee) · 4 Aug 2020

In this exploratory work, the authors attempt to assess the amount of ice nucleation active macromolecules that could be released from plants during, for example, a rainfall due to the washing effect of rainwater. This case study focuses on the cold-adapted tree Betula pendula (birch) that is known to produce ice-nucleating macromolecules. The long-term goal is to assess the extent of plants' contribution to the bulk of ice nucleation active particles of biological origin in the environment – in leaf litter in particular – and eventually in the atmosphere.

The methods and sampling design are adequate for this exploratory project and they

permit the authors to make approximations about the amount of ice nucleating particles that could potentially be released and if they are reasonable in comparison to the amounts captured in rainwater under birch trees. Based on the way that the information is presented, it is clear that the authors are not defending this as the last word on the subject, but rather as a first approximation, that opens the door for further investigations.

It is out of character for me to have so few criticisms of a manuscript in this area of research, but I feel that the authors have not over-sold the implications of their work and that the methods are appropriate and well-conducted. Furthermore, the introduction is very interesting and presents pertinent motivation for this work all while informing the reader that we are at the start of a new direction of investigations.

My one criticism concerns the details of the scenario that the authors propose for how these ice-nucleating macro-molecules enter the atmosphere. In line 289 the authors state "... Highlighting the possible pathways of INMs to be transported into the atmosphere during rainfall." Aside from this not being a complete sentence, this remark does not account for the general trend of downward flux of atmospheric particles during a rainfall. During rainfall the INM released by plants will most probably be washed to the ground and be incorporated into litter. Depending on their hydrophilicity, they might be washed into the soil and percolate into the groundwater, etc. Even if they remain in the litter, there will need to be sufficient turbulence at the ground level to move these INM's into the atmosphere from their situation under the canopy. I am not saying that this is not possible. Rather, I think that the authors need to add some details to their story to suggest a more plausible pathway of how the INM's will get into the atmosphere. This scenario will set the stage for the types of experiments that will need to be conducted in the future to fill in the gaps of knowledge.

SPECIFIC COMMENTS All specific comments concern spelling: Line 35: replace "regarding by "of" Line 186: replace "leave" by "leaf" Line 240: replace "indispensibly" by "indispensable" Table 1 title: replace "Information of" by "Information for" Figure 6

legend: replace "All rain samples collected are" by "All rain samples collected were"

---

## Referee Comment (RC2) · Anonymous Referee #2 · 28 Aug 2020

Plants are a source of ice nucleating particles found in the atmosphere. What fraction of emitted particles is synthesized by plants and what fraction is generated by microorganisms thriving on their surfaces is an open question. Another open question concerns the mechanisms by which rainfall aerosolises either kind of particle. Through the analysis of ice nucleating molecules (INMs) washed-off different parts of birch trees (in vivo) and in rain sampled below birch canopies and in open areas (in situ) the present study provides further proof that plants are sources of such entities released to the environment. Sampling and analysis were done very well. Results are clearly presented and the manuscript is overall a good read.

Of major concern to me are interpretations that are biased by the lower limit of detection being around 100'000 INMs cm-2, as gleaned from Figure 5. Though the design of the freezing assay provides for exploring freezing temperatures approaching homogenous freezing, it becomes increasingly 'blind' towards the warmer side of the temperature range because of the small sample volumes analysed. Not taking this fact into account leads to the questionable interpretation that the phytobiome on the surfaces of birch trees is a minor contributor to the population of INMs released to the environment (e.g. lines 229 to 231 and lines 242 to 243). Certainly true for temperatures below about -20 °C, this interpretation is most likely not true for temperatures above -10 °C or so. Support for this guess can be found in Figure 5, Trees D and G, where INM concentrations on leaves start to overtake those of other parts at around -17 °C. If data at warmer temperatures would be available, they would probably show increasingly larger ratios of leaf INM concentrations to those of wood or bark towards the warmer end of the temperature range. Therefore, I would suggest to either mention this possibility in the 'blind spot' of the assay, or to explicitly limit interpretation to temperatures below -20 °C.

Another issue that I would like the authors to address concerns the conclusion of '...similarities between in-vivo prepared extracts and in-situ sampled rainwater.' (line 291). It is not entirely clear to me. By looking at Figures 5 and 6, I see similarities in the shape of INM spectra for Trees G and H (leafless), but not for Trees E and F (with leaves). Latter have INM spectra in rainwater with approximately linear slopes on the log-scale, while the spectra of sampled material from these trees are mostly horizontal between -34 °C and -25 °C, then diving off towards warmer temperatures. I think the manuscript would benefit from additional discussion of similarities and dissimilarities of the INM spectra.

Minor comments

Please provide information on INM concentrations found or not found in the ultrapure water used in the assays (laboratory blank) and also give an estimate for the lower limit

of detection (around 100'000 INMs cm-2 ?).

Line 100: consider replacing 'pole testing drill' with 'increment borer', which is the correct technical term (see website of the instrument producer, http://www.haglofcg.com/index.php/en/products/instruments/survey/389-increment-borers).

Why did you choose to report K for-34 °C and not (also) for a warmer temperature? The -34 °C are so close to homogenous freezing, that the relevance of INMs at this temperature in what happens in nature seems questionable to me. Where K(-34 °C) values are mentioned, perhaps add in brackets also K values for a warmer temperature (e.g. -25 °C?).

Line 183: Please say what 'y' and 'x' stand for, including their units. If 'y' is K(-34 °C) in [cm-2] and 'x' is 'distance from surface of the stem' in [cm], then the concentration of INMs halves every 0.6 cm from the surface towards the core of the stem. Expressed this way, the information contained in the equation would be more amenable.

Line 190: '...samples did not show any ice nucleation activity,...' This statement depends on the detection limit of the freezing assay. To be more precise, you could say something like '...samples had K(-34 °C) values below about 10ˆ-5 cm-2,...'

Line 199: What do you mean with 'blank samples'? Samples of ultrapure water or precipitation samples collected in an open area, outside the influence of a tree crown?

Lines 220 and 221: '... concentration too low to be captured with our freezing assay.' Again, I think it is important to state in the methods section the lower limit of detection and re-iterate it in a context like in these lines. Ice propagates quickly in or around plants (e.g. Hacker and Neuner, 2008, https://doi.org/10.1657/1523-0430(07-077)[HACKER]2.0.CO;2). Hence, a single freezing event (i.e. INM) can affect the entire plant.

Lines 235 and 236: ' This could be either due to the sample collector been situated

too far from the tree, due to the interaction between rain and the tree's surface being insufficient, or that the western part of the tree exhibits fewer INMs.' I find the second assumption most convincing. Does rain typically come with westerly winds? If so, particles detached during a storm would mainly be found to the East of the tree.

Lines 244 and 245: One of the litter samples analysed by Schnell and Vali (1973) was re-analysed recently and, in addition to P. Syringae, further ice-nucleating species were identified in it (Vasebi et al., 2019, https://doi.org/10.5194/bg-16-1675-2019).

Lines 254 to 274: This is a courageous extrapolation! The number of INMs potentially released by trees is impressive. But, would these INMs not have to be lofted to the height of cirrus clouds to become activated? Although the INMs themselves are small, the question remains whether they are aerosolised as such or associated with larger particles? I think this issue needs attention in future studies. It would be useful to see the same extrapolation for INMs active at a warmer temperature (e.g. -25 °C ?).

---

## Author Comment (AC1) · 16 Sep 2020

We would like to thank Cindy Morris for her review and positive assessment of our manuscript, and we appreciate her comments and criticism, which are very helpful for improving the manuscript. The comments and our answers are listed below. The referee's comments are marked with blue letters, the responses of the authors are written in green.

In this exploratory work, the authors attempt to assess the amount of ice nucleation active macromolecules that could be released from plants during, for example, a rainfall due to the washing effect of rainwater. This case study focuses on the cold-adapted tree Betula pendula (birch) that is known to produce ice-nucleating macromolecules. The long-term goal is to assess the extent of plants' contribution to the bulk of ice nucleation active particles of biological origin in the environment – in leaf litter in particular– and eventually in the atmosphere. The methods and sampling design are adequate for this exploratory project and they permit the authors to make approximations about the amount of ice nucleating particles that could potentially be released and if they are reasonable in comparison to the amounts captured in rainwater under birch trees. Based on the way that the in-formation is presented, it is clear that the authors are not defending this as the last word on the subject, but rather as a first approximation, that opens the door for further investigations. It is out of character for me to have so few criticisms of a manuscript in this area of research, but I feel that the authors have not over-sold the implications of their work and that the methods are appropriate and well-conducted. Furthermore, the introduction is very interesting and presents pertinent motivation for this work all while informing the reader that we are at the start of a new direction of investigations.

My one criticism concerns the details of the scenario that the authors propose for how these ice-nucleating macro-molecules enter the atmosphere. In line 289 the authors state "... Highlighting the possible pathways of INMs to be transported into the atmosphere during rainfall." Aside from this not being a complete sentence, this remark does not account for the general trend of downward flux of atmospheric particles during a rainfall. During rainfall the INM released by plants will most probably be washed to the ground and be incorporated into litter. Depending on their hydrophilicity, they might be washed into the soil and percolate into the groundwater, etc. Even if they remain in the litter, there will need to be sufficient turbulence at the ground level to move these INM's into the atmosphere from their situation under the canopy. I am not saying that this is not possible. Rather, I think that the authors need to add some details to their story to suggest a more plausible pathway of how the INM's will get into the atmosphere. This scenario will set the stage for the types of experiments that will need to be conducted in the future to fill in the gaps of knowledge

Author's response:

Indeed, it is important to discuss possible pathways of INMs from the tree into the atmosphere more specific.

Therefore, we changed the incomplete sentence in line 312 "*Highlighting the possible pathways of INMs to be transported into the atmosphere during rainfall.*" and added our assumptions more clearly:

'*The exact pathway of INMs being transported from the trees into the atmosphere during rainfall is still not elucidated. One natural assumption would be that INMs are washed off the tree's surface and deposit on the ground surface (i.e. leaf litter or soil). Through strong winds at the ground level, INMs could be aerosolized through abrasive dislodgment and transported further. Another pathway would be that the aqueous INM extracts on the ground form a liquid film and get aerosolized during the mechanical impact of following rain droplets, similar to the bioaerosol generation mechanism suggested by Joung et al. (2017); Wang et al. (2016); Kim et al. (2020). Thereafter, splash induced aerosol can be transported further during turbulent wind events and convection.*'

SPECIFIC COMMENTS:

All specific comments concern spelling:

Line 35: replace "re-garding by "of"

Line 186:  replace "leave" by "leaf"

Line 240:  replace "indispensibly"by "indispensable"

Table 1 title:  replace "Information of" by "Information for"

Figure 6 legend: replace "All rain samples collected are" by "All rain samples collected were"

Author's response:

We thank the referee for her remarks and changed the spelling accordingly in the manuscript.

**References:**

Joung, Y. S., Ge, Z., and Buie, C. R.: Bioaerosol generation by raindrops on soil, Nature Communications, 8, 14668, https://doi.org/10.1038/ncomms14668, 2017.

Kim, S., Wu, Z., Esmaili, E., Dombroskie, J. J., and Jung, S.: How a raindrop gets shattered on biological surfaces, Proceedings of the National Academy of Sciences of the United States of America, 117, 13901-13907, https://doi.org/10.1073/pnas.2002924117 2020.

Wang, B., Harder, T. H., Kelly, S. T., Piens, D. S., China, S., Kovarik, L., Keiluweit, M., Arey, B. W., Gilles, M. K., and Laskin, A.: Airborne soil organic particles generated by precipitation, Nature Geoscience, 9, 433-437, https://10.1038/ngeo2705, 2016.

---

## Author Comment (AC2) · 16 Sep 2020

We would like to thank #2 referee for the review of our manuscript, and we appreciate the comments and criticism, which we have been taken into account upon revision of our manuscript. The comments and our answers are listed below. The referee's comments are marked in blue, the authors' responses in green.

Plants are a source of ice nucleating particles found in the atmosphere. What fraction of emitted particles is synthesized by plants and what fraction is generated by microorganisms thriving on their surfaces is an open question. Another open question concerns the mechanisms by which rainfall aerosolises either kind of particle. Through the analysis of ice nucleating molecules (INMs) washed-off different parts of birch trees (in vivo) and in rain sampled below birch canopies and in open areas (in situ) the present study provides further proof that plants are sources of such entities released to the environment. Sampling and analysis were done very well. Results are clearly presented and the manuscript is overall a good read.

Of major concern to me are interpretations that are biased by the lower limit of detection being around 100'000 INMs cm-2, as gleaned from Figure 5. Though the design of the freezing assay provides for exploring freezing temperatures approaching homogenous freezing, it becomes increasingly 'blind' towards the warmer side of the temperature range because of the small sample volumes analysed. Not taking this fact into account leads to the questionable interpretation that the phytobiome on the surfaces of birch trees is a minor contributor to the population of INMs released to the environment (e.g. lines 229 to 231 and lines 242 to 243). Certainly true for temperatures below about -20∘C, this interpretation is most likely not true for temperatures above -10∘C or so. Support for this guess can be found in Figure 5, Trees D and G, where INM concentrations on leaves start to overtake those of other parts at around -17∘C. If data at warmer temperatures would be available, they would probably show increasingly larger ratios of leaf INM concentrations to those of wood or bark towards the warmer end of the temperature range. Therefore, I would suggest to either mention this possibility in the 'blind spot' of the assay, or to explicitly limit interpretation to temperatures below -20∘C.

Author's response:

We thank the referee for this very important remark. We agree that the volume of the droplets within our freezing assay is rather low (40 μm diameter corresponds to 34 pL spherical volume) compared to other systems e.g. TINA (Kunert et al., 2018), however, it is comparable to atmospheric droplets and allows to monitor heterogeneous ice nucleation down to -34°C. Nonetheless, as the referee pointed out, the low amount of volume limits us to see low concentrations of INPs, especially in the warmer regions. We added the detection limit of our method in the chapter 'Methods' and supplemented the text in lines 143-144 with the following sentence: '*To observe at least one heterogeneous freezing event within 100 droplets (total amount of droplets in one measurement), the lower detection limit of VODCA is about $10^3$ INMs/μL ($10^5$ INMs/cm$^2$).*' Even though the limit of detection (LOD) is relatively high, we observe ice

nucleation at temperatures warmer than -15°C, which underlines the fact that in our field experiments the number of ice nuclei were rather high.

To determine whether the phytobiome of the tree's surface contributes to heterogenous ice nucleation, we attempt to use a freezing assay with larger volumes in future studies. We deleted '*and the phytobiome on the surface of birch trees (including ice-nucleating bacteria) is a minor contributor*' in line 240. Furthermore, we clarified in line 265 that we did not observe high concentrations of ice-nucleating bacteria on our leave samples.

Another issue that I would like the authors to address concerns the conclusion of '...similarities between in-vivo prepared extracts and in-situ sampled rainwater.' (line 291). It is not entirely clear to me. By looking at Figures 5 and 6, I see similarities in the shape of INM spectra for Trees G and H (leafless), but not for Trees E and F (with leaves). Latter have INM spectra in rainwater with approximately linear slopes on the log-scale, while the spectra of sampled material from these trees are mostly horizontal between-34∘C and -25∘C, then diving off towards warmer temperatures. I think the manuscript would benefit from additional discussion of similarities and dissimilarities of the INM spectra.

Author's response:

We thank the referee for this comment. We delved deeper into the similarities and dissimilarities of *in-situ* (extracts) and *in-vivo* (rain) samples regarding their ice nucleation. To compare the two data sets we focused on the INM concentrations $K(-25°C)$ and $K(-34°C)$, the freezing on-set temperature ($T_{on}$) and the shapes of the freezing curves. We added the following sentences in the discussion in line 253-263.

*'When comparing the ice nucleation data of in-vivo prepared extracts to in-situ sampled rainwater, INM concentrations (both $K(-25°C)$ and $K(-34°C)$ values) of all samples analysed are considerably similar. Furthermore, the $T_{on}$ of the trees with leaves are comparable (Tree E: $T_{on}$, -15.0°C (in-vivo), -15.6°C (in-situ); Tree F: $T_{on}$, -17.6°C (in-vivo), -17.2°C (in-situ)). In contrast, $T_{on}$ of trees with no leaves vary more widely, with in-vivo extracts starting at higher freezing temperatures. Moreover, the shape of the cumulative nucleus spectra of trees with leaves (Tree E and F) differ. The curves of in-vivo samples from these trees initially ascend strongly and then flatten towards colder temperatures. In contrast, the spectra of the in-situ sample from Tree E is convex towards the y-axis in the beginning and changes then further into a linear increase. The curves of the in-situ samples from Tree F look rather linear. On the contrary, the cumulative nucleus spectra of in-situ and in-vivo samples from leafless trees (Tree G and H) have very similar shapes. Both, in-vivo and in-situ sample curves of Tree G increase sharply in the beginning and flatten towards colder temperatures. The in-vivo and in-situ curves from Tree H have rather linear slopes on the logarithmic scale.'*

In addition, we added $K(-25°C)$ values in the chapter 'Results'. In chapter 3.2., we modified the first sentence (line 190) to the following: *'Nearly all analysed surface extracts of bark, leaf and branch wood samples release INMs with $K(-25°C)$ and $K(-$*

*34°C) values in the order of magnitude between $10^5$ and $10^9$ cm$^{-2}$'.* Furthermore, we added in line 195-197: '*Within the samples from the same tree K(-25°C) as well as K(-34°C) values are considerably similar from Tree A, D and G. Comparing K(-25°C) to K(-34°C) values from all leaf samples, the concentrations increase towards warmer temperatures.*' Furthermore, we included *K(-25°C)* values in chapter 3.3. We changed the sentence in line 207: '*K(-25°C) and K(-34°C) values, calculated using Equation (3), varied in the order of magnitude between $10^5$ and $10^9$ per cm$^2$ (i.e. $10^5$ to $10^9$ INMs were extracted per cm$^2$ area of rain) and $10^6$ - $10^9$ cm$^{-2}$.*' and two sentences in line 213 and 214: '*On the one hand, all samples from leaf-covered trees exhibit similar K(-25°C) and K(-34°C) values (higher than $10^7$ INM cm$^{-2}$). On the other hand, samples from trees without leaves show a higher variation.*'

Furthermore, in the 'Conclusion' we changed the sentences in line 319 to the following: '*We found high similarities regarding the INM concentrations between in-vivo prepared extracts and in-situ sampled rainwater,[…]*'.

Minor comments

Please provide information on INM concentrations found or not found in the ultrapure water used in the assays (laboratory blank) and also give an estimate for the lower limit of detection (around 100'000 INMs cm-2 ?).

Author's response:

We will provide the freezing curve of ultra-pure water (produced with Millipore® SAS SIMSV001, Merck Millipore, USA) in the supplement. We added the following in chapter 2.4., line 140: '*Prior to sample measurements, a reference sample (ultra-pure water produced with Millipore® SAS SIMSV001, Merck Millipore, USA) was analysed (see Figure 8 in Supplementary).*'

[Figure]

*Figure 8: Ice nucleation curve of ultra-pure water (3 measurements), recorded with VODCA.*

We added the detection limit of our method in chapter 'Methods' and supplemented the text in lines 143-144 with the following sentence: '*To observe at least one heterogeneous freezing event within 100 droplets (total amount of droplets in one measurement), the lower detection limit of VODCA is about $10^3$ INMs/µL ($10^5$ INMs/cm$^2$).*'

Line 100: consider replacing 'pole testing drill' with 'increment borer', which is the correct technical term (see website of the instrument producer, http://www.haglofcg.com/index.php/en/products/instruments/survey/389-increment-borers.

Author's response:

We thank the referee for pointing this out and changed the wording to 'increment borer'.

Why did you choose to report K for-34∘C and not (also) for a warmer temperature? The -34∘C are so close to homogenous freezing, that the relevance of INMs at this temperature in what happens in nature seems questionable to me. Where K(-34∘C) values are mentioned, perhaps add in brackets also K values for a warmer temperature (e.g. -25∘C?).

Author's response:

We thank the referee for this comment. We decided to point out *K(-34°C)* values since it includes all heterogenous freezing events using our assay (homogenous freezing starts at -35°C). However, to improve our results we included *K(-25°C)* in the paper.

Line 183: Please say what 'y' and 'x' stand for, including their units. If 'y' is K(-34∘C)in [cm-2] and 'x' is 'distance from surface of the stem' in [cm], then the concentration of INMs halves every 0.6 cm from the surface towards the core of the stem. Expressed this way, the information contained in the equation would be more amenable.

Author's response:

Indeed, y describes *K(-34°C)* [cm$^{-2}$] and x the depth [cm]. We included the description with the units in the text (line 186).

Line 190: '...samples did not show any ice nucleation activity,...' This statement depends on the detection limit of the freezing assay. To be more precise, you could say something like '...samples had K(-34∘C) values below about 10ˆ-5 cm-2,...'

Author's response:

The authors thank the referee for this comment and we changed the sentence accordingly: '*Two of all analysed samples had K(-34°C) values below the limit of detection, both of which were primary wood, […]*'.

Line 199: What do you mean with 'blank samples'? Samples of ultrapure water or precipitation samples collected in an open area, outside the influence of a tree crown?

Author's response:

Blank samples correspond to the sample holder which was put not directly under the canopy as we did for the 'samples' but away from the tree where it stood in open terrain (no interaction between rain and tree). We added the following in the text (line 206): '*[…] blank samples (pure rainwater, collectors were set up next to the trees in open terrain) […]*'.

Lines 220 and 221: '... concentration too low to be captured with our freezing assay.' Again, I think it is important to state in the methods section the lower limit of detection and reiterate it in a context like in these lines. Ice propagates quickly in or around plants (e.g. Hacker and Neuner, 2008, https://doi.org/10.1657/1523-0430(07-077)[HACKER]2.0.CO;2). Hence, a single freezing event (i.e. INM) can affect the entire plant.

Author's response:

As mentioned above, we added the LOD in chapter 'Methods'. In addition, we added the following in our manuscript (line 229,230): '*Nonetheless, when larger water droplets are on the surface of a plant tissue, even a small concentration of INMs could be enough to induce one ice nucleation event, leading to ice propagation and affecting the entire plant (Hacker and Neuner, 2008).*'

Lines 235 and 236: ' This could be either due to the sample collector been situated too far from the tree, due to the interaction between rain and the tree's surface being insufficient, or that the western part of the tree exhibits fewer INMs.' I find the second assumption most convincing. Does rain typically come with westerly winds? If so, particles detached during a storm would mainly be found to the East of the tree.

Author's response:

In general, precipitation comes with westerly winds in the region where we took the samples. On the weather side of the trees considerably more mosses and lichens grow which leads to an increased microbiological activity.

The west sample from Tree G had lower INM concentrations than the other two samples from this tree (S, NE). Indeed, one explanation could be that INMs were detached during a storm and would mainly be found to the east of the tree. However, we did not observe the same trend in the other trees. Thus, to clarify this hypothesis more data is needed.

Lines 244 and 245: One of the litter samples analysed by Schnell and Vali (1973) was reanalysed recently and, in addition to P. Syringae, further ice-nucleating species were identified in it (Vasebi et al., 2019, https://doi.org/10.5194/bg-16-1675-2019).

Author's response:

We have the reference in the 'Introduction' (line 51). Furthermore, we added the reference in the 'Discussion' and added the following sentence in lines 267-268: '*Recently, the leaf litter was re-analysed after almost 50 years and a variety of species inducing ice nucleation were characterized.*'

Lines 254 to 274: This is a courageous extrapolation! The number of INMs potentially released by trees is impressive. But, would these INMs not have to be lofted to the height of cirrus clouds to become activated? Although the INMs themselves are small, the question remains whether they are aerosolised as such or associated with larger particles? I think this issue needs attention in future studies. It would be useful to see the same extrapolation for INMs active at a warmer temperature (e.g. -25∘C ?).

Author's response:

We thank the referee for this comment. The authors agree that further studies are needed in order to investigate the transport of biological INMs and INPs from the land surface to high altitudes (vertical profiles). The question how INMs from the trees are getting aerosolised is still not answered. On the basis of the comment from #1 referee, Cindy Morris we added our assumptions on the transport of INMs more clearly in the 'Conclusion' (lines 312-318):

*'The exact pathway of INMs being transported from the trees into the atmosphere during rainfall is still not elucidated. One natural assumption would be that INMs are washed off the tree's surface and deposit on the ground surface (i.e. leaf litter or soil). Through strong winds at the ground level, INMs could be aerosolized through abrasive dislodgment and transported further. Another pathway would be that the aqueous INM extracts on the ground form a liquid film and get aerosolized during the mechanical impact of following rain droplets, similar to the bioaerosol generation mechanism suggested by Joung et al. (2017);Wang et al. (2016);(Kim et al., 2020). Thereafter, splash induced aerosol can be transported further during turbulent wind events and convection.*'

**References:**

Hacker, J., and Neuner, G.: Ice Propagation in Dehardened Alpine Plant Species Studied by Infrared Differential Thermal Analysis (IDTA), Arctic, Antarctic, and Alpine Research, 40, 660-670, 611, https://doi.org/10.1657/1523-0430(07-077), 2008.

Joung, Y. S., Ge, Z., and Buie, C. R.: Bioaerosol generation by raindrops on soil, Nature Communications, 8, 14668, https://doi.org/10.1038/ncomms14668, 2017.

Kim, S., Wu, Z., Esmaili, E., Dombroskie, J. J., and Jung, S.: How a raindrop gets shattered on biological surfaces, Proceedings of the National Academy of Sciences of the United States of America, 117, 13901-13907, https://doi.org/10.1073/pnas.2002924117 2020.

Kunert, A. T., Lamneck, M., Helleis, F., Pöschl, U., Pöhlker, M. L., and Fröhlich-Nowoisky, J.: Twin-plate Ice Nucleation Assay (TINA) with infrared detection for high-throughput droplet freezing experiments with biological ice nuclei in laboratory and field samples, Atmospheric Measurement Techniques, 11, https://doi.org/10.5194/amt-11-6327-2018, 2018.

Wang, B., Harder, T. H., Kelly, S. T., Piens, D. S., China, S., Kovarik, L., Keiluweit, M., Arey, B. W., Gilles, M. K., and Laskin, A.: Airborne soil organic particles generated by precipitation, Nature Geoscience, 9, 433-437, https://10.1038/ngeo2705, 2016.